# Structured Semantic Modeling of Scientific Citation Intents

Roger Ferrod, Luigi Di Caro (✉), and Claudio Schifanella

Department of Computer Science, University of Turin
{roger.ferrod,luigi.dicaro,claudio.schifanella}@unito.it

**Abstract.** The search for relevant information within large scholarly databases is becoming an unaffordable task where deeper semantic representations of citations could give impactful contributions. While some researchers have already proposed models and categories of citations, this often remains at a theoretical level only or it simply reduces the problem to a short-text classification of the context sentence. In this work, we propose *CiTelling*: a radically new model of fine-grained semantic structures lying behind citational sentences able to represent their intent and features. After an extensive and multiple annotation of 1380 citations[1], we tested the validity and the reliability of the proposal through both qualitative and quantitative analyses. In particular, we were able to 1) extend the current depth of existing semantic representations when used in computational scenarios, 2) achieve high inter-annotator agreement and 3) obtain state-of-the-art classification results with straightforward neural network models.

**Keywords:** Citation Semantics · Scientific Literature Exploration · Semantic Annotation

## 1 Introduction

Exploring and understanding the heart of millions of scientific articles is not an easy situation for a young researcher. Actually, only keeping abreast of research progress is becoming an increasingly difficult task even for senior and experienced scientists. Digital technologies are now being adopted since years to lighten such process, by providing advanced "*semantic*" search services based on keywords rather than metadata extraction and filtering procedures.

Numerous techniques have been developed to analyze large amounts of data such as the petabytes produced by the Large Hadron Collider or the hundreds of millions of bases contained in the human genome, but natural language cannot be naturally represented by numbers and easily manipulated by computers. Moreover, the research literature is made of complex textual content which is naturally oriented to be read by humans only.

However, while the actual understanding of the scientific content remains in the researchers' hands, a great support may come from the application of data

---

[1] https://github.com/rogerferrod/CiTelling

and language technologies to the citational aspect of the articles. Citations, indeed, represent a fundamental mechanism for both making new research and keeping track of what is going on within a scientific field. For example, an article $A$ may cite an article $B$ for different purposes: to extend, to criticize, to compare with, to refer to some used data or technique, and so forth. In other words, citations are crucial for both production and search, but semantic technologies are still far away from being supportive to the daily work of researchers.

Under this light, models such as [22] investigated possible types of citations, together with descriptions and examples. However, categories are sometimes very specific and linked to a few examples, making them difficult to be employed in concrete applications. Alternatively, recent works proposed Machine Learning approaches for the automatic classification of sentences containing citations into few classes such as *extension*, *use*, etc. The problem with the latter approach is the reduction of the complexity of the single citation semantics into a short-text classification task, where the expected output is simply one label to associate with the sentence containing the citation.

In this contribution, we draw from these experiences to create a fine-grained semantic model of citations which can be instead employed in computational scenarios, manually producing an annotation (available at `https://github.com/rogerferrod/CiTelling`) of more than one thousand cases for testing its validity through both human agreements and neural network-based classification results. The goal of this work is thus threefold: 1) to propose a computationally-affordable semantic model for citations inheriting and enriching key features of state-of-the-art efforts; 2) to provide an extensive and multiple-user annotation of citational sentences; and 3) to demonstrate the model validity through both human- and machine-based evaluations.

## 2   Related Work

There exists a large body of literature focusing on the processing of scientific texts for purposes such as data curation (e.g. [17]), search (e.g., [4]), topic modeling (e.g., [3]), summarization (e.g., [25]), and so forth. In this context, our work has similarities with different approaches related to the modeling and use of the citations within large scholar databases, such as *(i)* semantic modeling of citations (e.g., [22, 16, 6, 13]), *(ii)* data analysis and extraction of relevant information (e.g., [9, 20, 23]), and *(iii)* exploration of the scientific literature by means of faceted search queries and visualization tools (e.g., [10, 11, 2, 18, 1]).

In this paper, we focus on the first task of modeling citations, specifically inheriting both the theoretical and top-down approach of *CiTo* [22] and recent state-of-the-art technologies for automatic citation classification [6]. More in details, in [22], the authors identified and formalized different types of possible citation meanings in scientific articles. However, the proposed ontology includes a wide set of complex cases, making it exclusively suitable for manual (and costly) annotations of individual references. In [12], the authors presented an

unsupervised technique based instead on a completely automatic clustering process, identifying and describing 11 classes of citations.

More recently, [6] proposed a classifier based on Scaffolds models [24] that was able to identify 6 classes of citations on the ACL-ARC dataset [7] and 3 classes on a larger dataset named *SciCite* [6] with state-of-the-art accuracy levels. In particular, we used these works as baseline for the evaluation, finding that our model allows to achieve comparable performance with extremely simpler neural network-based classifiers on equally-distributed and semantically-deeper citation intents.

## 3 Motivations and Research Questions

### 3.1 Semantic structures in citations

Our main goal is to build a fine-grained semantic representation of citations to capture and harmonize their *i*) intent type, *ii*) direction, *iii*) objects or concepts involved, and *iv*) context. To better express our idea, let us consider the example below:

"*We use the Scaffold network classifier (Cohan 2019) to incorporate syntactic structures [...]*"

According to *CiTo* [22], this example should fit the ontological category *use*, as well as for the state-of-the-art classification system proposed in [6]. However, in the latter case the procedure is only limited to the classification of the sentence.

In our work, instead, we face the citation classification problem under a more structured semantic view. In particular, our aim is to model the above citation in the following way:

```
(a)          SUBJECT_PAPER_ID: <this_paper_id>;
             INTENT: <uses>;
             OBJECT: <scaffold network classifier>;
             OF_PAPER_ID: <Cohan_2019_id>;
             IN_CONTEXT: <to incorporate syntactic structures>;
```

A part from being more informative than in [22] and [6], our model is able to cope with more complex (but frequent) cases. For instance, consider the citation below:

"*(Peter et al 2018) uses the SVD factorization method.*"

In this second case, both [22] and [6] approaches would simply associate a label *use* to the whole sentence, as in the previous example. Instead, we model this different case as:

```
(b)              SUBJECT_PAPER_ID: <Peter_2018_ID>;
                 INTENT: <uses>;
                 OBJECT: <SVD>;
                 OF_PAPER_ID: n/a;
                 CONTEXT: n/a;
```

It is important to note that, in the latter case *(b)*, the $<this\_paper\_id>$ identifier does not enter into the model, since the paper only plays the role of *container*. In other words, $<this\_paper\_id>$ only contains the $<uses>$ information that links the cited paper $<Peter\_2018\_ID>$ with the object $<SVD>$. To the best of our knowledge, this is the first attempt to extract source-agnostic knowledge from scholarly databases, as detailed in the next section.

### 3.2   Active and passive roles in citations

To better understand the advantages of our proposal, one can think at the related work section of a scientific article, which usually expresses definitions, facts and comparative analyses of existing works in the literature. Such section is indeed an extremely rich source of information to model knowledge related to external articles. Usually, semantic analyses of scholarly articles are focused on the modeling of their direct content, whereas they usually contain knowledge about (mentioned) existing works.

A part from the citation class, we further model citational sentences through *active* and *passive* roles. The first case includes a relation of a certain class/intent between a source paper $A$ and a referenced paper $B$, with a focus on some research objects in $B$. In the second case, the relationship lies between a referenced paper $B$ (mentioned by the citing paper $A$) and some research object presented by an unknown third-party paper. In this second situation, it is $B$ that covers the role of subject, proposing, adding or using the object. In other words, the source paper $A$ does not cover any semantic role of interest (while it simply functions as a container for the mentioned citation). Please note that, in the previous Example (b), it would be a mistake to classify the *A-B* relation with the "*use*" class as currently done by current approaches, since it is not $A$ that uses the mentioned research object. To the best of our knowledge, this represents the first attempt to model structured semantics behind citations, as well as such subject-oriented role. Hereafter we distinguish the two roles by calling them *A-subject* and *B-subject* depending on who holds the role of subject.

In Figure 1 we illustrate a comparison between current models and our proposal. In particular, *CiTelling* embodies information about roles and relationships that involve fine-grained objects rather than the whole papers. Notice that, while existing state-of-the-art computational models face the problem as a simple short-text classification task, we consider the citation semantics under a semantically richer and structured view. This creates a *knowledge graph* instead of a simpler (labeled) network of articles.

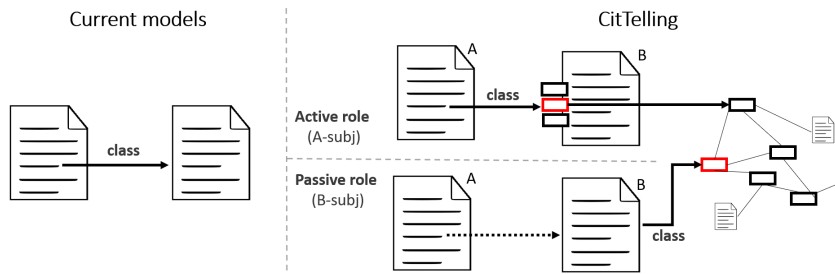

**Fig. 1.** Comparison of existing models (on the left) with our proposal (right). While the relations of current models involve two papers as a whole (associating an intent with each citation), *CiTelling* is able to highlight particular topics of the cited text (objects marked in red), separating the active from the passive role cases.

## 4    Semantic Structuring of Citation Intents

Since the scaffold network model proposed in [6] represents the state of the art on citation classification using the ACL-ARC dataset (developed in [7]), we used its classes as starting point. The original labels were the following: *background*, *extends*, *uses*, *motivation*, *compare/contrast*, and *future work*. Then, we integrated other two classes from the results of [12]: *propose* and *analyze*. Finally, we added the more formal (and rare) *CiTo* proposals: *critiques* and *data source*. After a careful analysis of their labeling and meaning, we ended up with the five intents (or classes) we thought to be more informative, as shown in Table 1. For example, we decided to exclude the intent *background* as it is often used as a generic "relatedness-based" container.

**Table 1.** Proposed model of citation intents, integrating features from existing models. \**Analyze* is derived from the most specific *Report* label presented in [12].

| Intent | Optional subclass | CiTo [22] | SciSite [6] | CitExp [12] |
|---|---|---|---|---|
| *Proposes* | | | | x |
| *Uses* | [dataset] | x | x | |
| *Extends* | | x | x | |
| *Analyzes* | [critiques] | | | x* |
| *Compares* | [contrasts] | | x | |

Taking inspiration from the *CiTo* ontology, which contains more specific categories such as "*usesDataFrom*" and "*usesConclusionsFrom*", we decided to also consider differences within the intents *uses*, *compares* and *analyzes*. We have therefore added subcategories to highlight the use of *datasets* or *results* in the *use* class and to distinguish dissimilarity (i.e. *contrast*) in *compare*. In the same way, it is useful to capture the negative analyses, which highlight critical issues

related to the citation in the *analyze* class; this latter case is extremely rare but very informative, thus we finally opted for its inclusion in our model.

### 4.1  Object and Context fields

Another innovative point of our contribution, compared to previous works, is the introduction of two semantic fields *object* and *context* for further modeling the citational semantics, as illustrated in Figure 2. In this section, we present definitions and examples for these two fields.

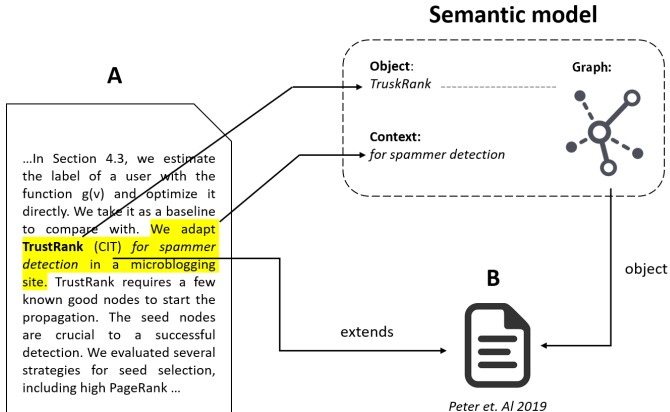

**Fig. 2.** General overview of the proposed semantic model.

With *object*, we mean a mandatory concept taken into consideration by the citation, whose meaning changes according to the class. For *context*, we mean an optional additional background information, or constraints, which can be useful to disambiguate the *object*. For example, in the sentence:

> "*Adaptive modulation techniques over Nakagami-m fading channels were also investigated in [CIT] for mobile wireless channel.*"

the *object* would be "*adaptive modulation techniques*", which is contextualized by "*for mobile wireless channel*".

More in detail, the *object* can be defined as the minimal span of text to which the quotation refers (e.g. the extended or analyzed concept). Consequently, it contains the smallest amount of information able to identify the concept of interest. The *context* is instead represented by a usually larger text surrounding the *object* helping disambiguate or specify it. A direct application of this model could be within a semantic search engine, where the *context* may accompany the object through a tool-tip or other visualization tools, providing additional information on the purpose of the *object* (e.g., "*for matrix factorization*") rather

than on the domain of application (e.g., "*in elliptic curve cryptography*"). Additionally, the *context* can be used for integrating additional features into the encoding of *object* meanings (for example, through the employment of recent BERT-like context-dependent embeddings [8]).

Note that we modeled the *object* and the *context* fields in a purely semantic way, avoiding to create lexical-semantic interfaces based on part-of-speech tags, patterns or syntactic structures. While this choice leaves higher margin to subjective evaluations, our aim was twofold: $i$) to deeply evaluate the model through inter-annotation agreements in absence of physical/narrow lexical constraints; and $ii$) to leave room for future automated labeling technologies (e.g. Ontology Learning, Transformer-based Machine Learning, etc.).

### 4.2    Definition of the citation intents

In this section, we provide definitions and examples for the chosen set of intents (or classes) used in our model.

With the exceptions of *propose*[2] and *compare*, all the other classes may be associated with the two roles previously described in Section 3.2. For this reason, the formalization that follows will consider both cases. It is possible to generalize the relationships in the following way:

$$A \xrightarrow{\text{class-label}} \overset{\text{[A-subject]}}{object} \xleftarrow{\text{proposes}} B$$

$$B \xrightarrow{\text{class-label}} \overset{\text{[B-subject]}}{object}$$

Hereafter we indicate with $A$ the citing paper (i.e. the paper under analysis) and with $B$ the cited paper. Please note that the *propose* relation is implicit in the *A-subject* representation, since the role of $B$ is that of containing *object*.

***Propose*** **class.** This class models a particular contribution (i.e. *object*) of a paper. More formally:

$$\exists\ B,\ object\ |\ B \xrightarrow{\text{proposes}} object$$

In words, there exists an article $B$ and one contained concept *object* such that $B$ proposes *object*. Different examples of *propose* citations are shown below:

---

[2] *propose* cannot have a relation of type *A-subject* since if the paper $A$ simply proposes *object*, then there is no reason to quote another paper $B$. Different is the case in which the authors of a paper $A$ propose an *object* referring to a paper $B$, but in that case the correct relationship is of type *B-subject*.

(example 1) "The $\underset{object}{\underline{relational\ model}}$ was first introduced in the work by Codd [CIT]."

(example 2) "$\underset{object}{\underline{Anisotropic\ diffusion}}$ was proposed $\underset{context}{\underline{in\ the\ context\ of\ scale\ space}}$ [CIT]."

**Use class** This class models the simple use of some object, citing an external article B. More formally:

[A-subject]
$\exists\ A,\ B,\ object\ |\ A \xrightarrow{\text{uses}} B.object$

In words, there exists an article $B$ cited within article $A$ where $B.object$ is used in $A$.

[B-subject]
$\exists\ B,\ object\ |\ B \xrightarrow{\text{uses}} object$

In this case, there exists an article $B$ cited within article $A$ such that $B$ uses some *object* (thus, $A$ plays only the role of article-container). Different examples of *use* citations are shown below:

(A-subject example - dataset) "$\underset{context}{\underline{For\ faces}}$, we used $\underset{object}{\underline{FaceScrub\ dataset}}$ from [CIT]."

(B-subject example) "This $\underset{object}{\underline{MMSE\ representation}}$ was used in [CIT], $\underset{context}{\underline{to\ prove\ the\ EPI}}$."

**Extend class** This citation intent models the natural process of scientific evolution, that is the possibility of a paper to modify (adapting or enriching) another work. More formally:

[A-subject]
$\exists\ A,\ B,\ object\ |\ A \xrightarrow{\text{extends}} B.object$

In words, there exists an article $B$ cited within article $A$ where $B.object$ is extended in $A$.

*[B-subject]*
$\exists\ B,\ object\ |\ B \xrightarrow{\text{extends}} object$

In this case, there exists an article $B$ cited within article $A$ such that $B$ extends some *object*. For example:

> (A-subject) "*This algorithm is a generalization of the famous* min-norm point algorithm [CIT]."
> object

> (B-subject) "*Ernst et al. [CIT] tackle the static ASP by using a specialized* simplex algorithm *for the single runway case and* object
> extend it to the multiple runway case."
> context

It is important to notice the difference between *extend* and *use*. In particular, extending a work means its use after the application of some changes. This is why cases such as "*Our work is based on CIT*" and "*Following the work of CIT we [...]*" can be also considered instances of the *extend* class.

**Analyze class** This citation type identifies processes of analysis and discussion on specific topics. More formally:

*[A-subject]*
$\exists\ A,\ B,\ object\ |\ A \xrightarrow{\text{analyzes}} B.object$

*[B-subject]*
$\exists\ B,\ object\ |\ B \xrightarrow{\text{analyzes}} object$

Examples of *analyze* citations are shown below:

> (A-subject example) "*We have conducted a survey to discuss* Big Data Frameworks [CIT]."
> object

> (B-subject example - critique) "*Refer to CIT for a discussion on the* optical design *problem of* HMDs."
> context        object

***Compare* class** This class identifies similarities or contrasts between articles over a specific research object. More formally:

$$\exists\ A,\ B,\ object \mid A.object \xleftrightarrow{\text{compares}} B.object$$

This particular relationship is symmetric since the *object* can be either in A or B. Examples of this type of citations are shown below:

> (example 1) "*For VGG, our $\underset{object}{\underline{latency}}$ is longer than [CIT] due to 45% frequency gap.*"

> (example 2) "*This approach is similar to the $\underset{object}{\underline{strategy}}$ defined in [CIT] $\underset{context}{\underline{as\ gap\ recovery}}$.*"

Notice that, in the first case, *latency* is shared between A and B, meanwhile in the second example there are two different words (*approach* belonging to A, and *strategy* to B) referring to the same concept.

### 4.3   Data Selection and Annotation

On such modeling basis, we built a balanced dataset with 276 instances for each class, for a total of 1380 instances. As already mentioned, in contrast with the state of the art baselines, we avoided to consider a *background* label since it usually represents a generic class that collects all citations "escaped" from any meaningful classification.

We have randomly sampled 10K papers from the Semantic Scholar corpus[3] obtaining, through ParsCit [15], more than 200K citations. For simplicity, we filtered out the sentences longer than 40 words; however in this way we captured most of the cases. Then, the selection of the candidate citational sentences has been carried out through a first phase of random sampling over such extracted citations, for each class. This process was based on different techniques: for classes such as *use* and *extend* we made use of the classifier provided in [6], while for others (*analyze* and *propose*) we employed a keyword-based random search. The candidate citational sentences were then cleaned out of the noise with a manual validation.

The second phase regarded the annotation of the sentences with the proposed structured semantic model. Three different annotators (the authors of the present paper) separately validated the class label and annotated the *A-Subject* vs *B-Subject* role, the *object* and the *context* fields for each instance. At the same time, the sub-classification operations were carried out, highlighting the subtypes of the classes (e.g. use:data/use:other and analyze:analyze/analyze:critique).

---

[3] https://www.semanticscholar.org/

The results of the inter-annotation agreement, calculated through the use of the Bleu score [19], are shown in Figure 4.

In performing this operation we found that there may exist some overlap among the classes and therefore it was useful to set up a disambiguation mechanism. For example, the type *extend* is considered more informative than *propose*. More in detail, we defined an ordered list of the classes to guide the disambiguation of ambiguous cases:

$$extend > analyze > compare > use > propose$$

In this way, it is possible to disambiguate sentences like: "*We use an extension of [CIT]*" (classified as *extend* instead of *use*) and "*We propose a comparative analysis beetween SVC [CIT] and NuSVC*" (labeled with *analyze*).

## 5    Evaluation

In this section we first report an analysis of the annotation task together with an evaluation of the impact of intent roles (active/passive), intent subclasses (see Table 1), and the obtained inter-annotation agreement. Secondly, we employed the model (through the annotated dataset) in a downstream task, i.e., intent classification, to be able to make comparisons with the current state of the art.

### 5.1    Roles, Subclasses and Inter-annotation Agreement

After the annotation phase, we have identified a clear diversification in the distribution of roles, as shown in Figure 3. The subclasses are distributed as follows: 10.14% of the instances in the *use* class are further labeled as "*use data/results*", meanwhile 15.22% of the *compare* class instances were better specified as "*contrast*". Finally, the rarest, 6.16% of the *analyze* instances are of subtype "*critique/error*". These statistics suggest the utility of sub-classifying the intents, in order to preserve useful information which can be further processed, analyzed and exploited for automatic classification and reasoning purposes.

Another interesting consideration concerns the nature of the *object* in the various citation intents: classes such as *extend* and *compare* are mostly associated with very generic objects like *method*, *approach*, *work* or *study* (respectively 39% and 31% of the total number of objects), meanwhile *use* and *analyze* mention very specific objects (a kind of named entities) such as *LSTM*, *Vertex-II Pro* or *CPLEX 12.6* (respectively 39% and 15% of the total).

To calculate the overall agreement on the three annotations of intents {*A1, A2, A3*}, we averaged the scores obtained from the three pairs <*A1,A2*>, <*A1,A3*>, and <*A2,A3*>. More in detail, for each pair we counted a +1 contribution if both annotators labeled the sentence with the same intent. Then, by dividing this value by the number of annotated citations we obtained a global averaged score which was particularly high (0.88).

Then, for the in-text semantic annotations, we computed Bleu scores [19] on *object* and *context*, reaching the scores of 0.78 and 0.55 respectively. These results

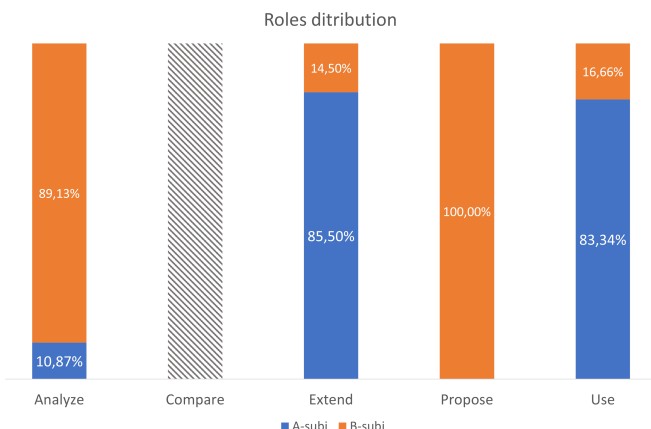

**Fig. 3.** Predominance of roles types (*A-subject* / *B-subject*) in the classes distribution. As described in Section 4.2, role types are not defined for the *Compare* class.

are in line with our initial expectations: since *context* has a less constrained definition with respect to *object*, it is more susceptible to lexical variations and different textual span interpretations. Moreover, *context* has an average length which is greater than that of *object* (5.77 words vs 2.40 words). The *object* field reached instead high agreement levels. The whole result set broken down by class is reported in Figure 4. We omitted the results of the agreement on roles as they correspond to an average score of 1.0 (the same considerations hold for the intent subclasses).

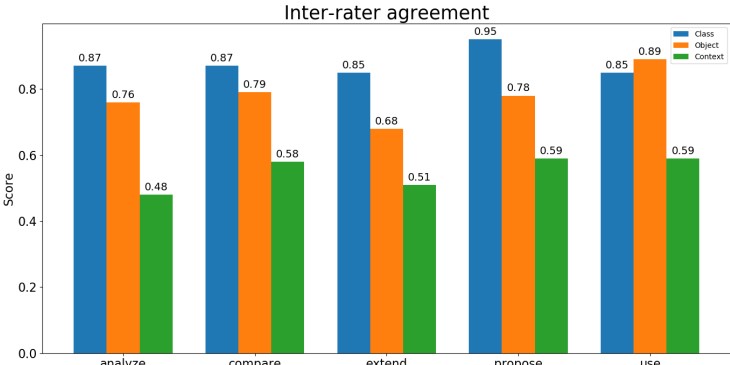

**Fig. 4.** Averaged inter-rater agreement among the three annotators for each citation intent. The *object* and *context* bars represent averaged inter-annotation Bleu scores.

Among the different citation intents, *propose* resulted to be very easy to classify by the annotators, with an agreement of 0.95. The *object* field goes from an average Bleu score of 0.68 for the *extend* intent to 0.89 for *use*. Then, the *context* field, as already stated, reached values between 0.48 and 0.59.

### 5.2   Downstream Task Evaluation

We evaluated our model by building a simple classifier on the annotated data, comparing the results with the existing state of the art. In particular, we have focused on the outcomes reported by [6]. Unlike the ACL-ARC (used by [7] and [6]) and SciCite [6] datasets, our dataset is balanced and does not include generic labels such as *background*. The distribution of the classes is shown in Table 2.

**Table 2.** Classes distribution and overlap among datasets.

| Intent | *ACL-ARC* (# 1941) | *SciCite* (# 11020) | *CiTelling* (# 1380) |
|---|---|---|---|
| Method | - | 29% | - |
| Result comparison | - | 13% | - |
| Background | 51% | 58% | - |
| Future work | 4% | - | - |
| Motivation | 5% | - | - |
| Extend | 4% | - | 20% |
| Use | 19% | - | 20% |
| Compare | 18% | - | 20% |
| Propose | - | - | 20% |
| Analyze | - | - | 20% |

Moreover while the *SciCite* dataset contains more than 11k elements, *ACL-ARC* [7] has a number of instances (1941) comparable to our dataset (1380) and a similar number of classes, albeit with a completely different distribution.

In contrast with the state-of-the-art classifiers under comparison in this section, ours is based on an extremely simple architecture. This choice comes from the aim of evaluating the semantic coherence and power of the proposed *CiTelling* model by comparing it with the results of complex neural architectures applied on state-of-the-art citational representation models. In particular, we adopted a single biLSTM layer densely connected to a softmax function, using the citational sentence as unique input. For the initial word representation layer, we employed pretrained *fastText* word embeddings [14].

Despite the simplicity of the classifier, the results are in line with other more sophisticated existing architectures, in particular with the outcomes reported in [6]. Notice that these methods cannot be directly evaluated on our data since they require further input features and metadata such as section titles and citation markers. Contrariwise, it was possible to apply a simple neural architecture (biLSTM with optimized hyper-parameters[4]) on the existing ACL-ARC data,

---

[4] 50 input dim, 12 (x2) hidden units, dropout 0.7, L2 penalty 1e-06.

which reached a significantly low F1 score (38.0%) compared to what obtained by the same neural model on our *CiTelling* data (65.8%). Furthermore, we noticed more balanced values of Precision and Recall with respect to the compared approaches. An overview of the results is reported in Table 3.

Since our model also integrates directional information (i.e., active and passive roles), we further carried out an additional experimentation by training a neural network performing role classification. By using a *biGRU* architecture [5] with standard settings[5], we obtained a F1-score of 77.6%, with Precision and Recall of 80.6% and 74.8% respectively.

**Table 3.** Intent classification results on *CiTelling* and *ACL-ARC* data.

| *CiTelling* data | F1 Score | Precision | Recall |
|---|---|---|---|
| biLSTM | 65.8 | 66.2 | 66.1 |

| *ACL-ARC* data | F1 Score | Precision | Recall |
|---|---|---|---|
| biLSTM | 38.0 | 44.0 | 37.0 |
| biLSTM + attention | 51.5 | 53.7 | 50.6 |
| biLSTM + attention + elmo | 54.2 | 59.2 | 51.6 |
| [Jurgens et al 2018] | 54.6 | 64.9 | 49.9 |
| biLSTM + attention + scaffolds | 63.1 | 71.7 | 58.2 |
| SciCite classifier | 67.9 | 81.3 | 62.5 |

## 6    Conclusions and Future Work

In this paper, we proposed a new semantic representation for modeling citations within a corpus of scholarly articles. In particular, we took inspiration form both theoretical bases and current computational approaches to both propose a novel semantic model and to create a publicly-available annotated resource. In contrast with the existing approaches aiming at labeling citations with some predefined classes, we put forward a structured model integrating an ontological view of the referenced objects within the literature.

Future developments of this work may include the integration of further articles metadata (e.g., sections, timestamps, etc.) and the management of sentences with multiple types of citations. For example, in the sentence "*CIT used an extension of X but the results are not satisfactory*" the citation plays two roles: it uses an extension of the *object* (therefore categorizable as *extend* with *B-subj* role) while the results of this operation are criticized (*analyze* with *A-subj* role). Furthermore, in the first case the *object* is *X* while in the second case the *object* is *CIT*; thus they are two distinct citation intents that can be extracted separately, enriching in this way the knowledge model.

---

[5] 50 input dim, 25 (x2) hidden units, dropout 0.8, L2 penalty 1e-05.

In order to build a knowledge model capable of integrating and exploiting all the captured concepts, further effort may be spent in the generalization of the objects. This operation could be facilitated by the presence of the *context* field. The objects automatically extracted from the text can then be aligned with existing ontologies such as [21], a large-scale ontology of research mainly in the field of Computer Science.

Our contribution will also enable the construction of a directed citation semantic graph which can be used for advanced analyses (e.g., graph embeddings) rather than semantic web search applications. For example, we can hypothesize the analysis of the knowledge graph with centrality measures, community detection algorithms or temporal analysis, in order to trace the evolution of communities and topics within specific citational paths.

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
