# OpenReview forum: "Structured Semantic Modeling of Scientific Citation Intents"
_eswc-conferences.org/ESWC/2021/Conference/Research_Track — ESWC 2021 Research_

### Official Review · AnonReviewer2 · 2021-01-13
**An inadequate attempt to establish a new model for scientific citation**

**Confidence:** 4
**Impact:** 1
**Design And Technical Quality:** 1

**Review:**

* The answers in the rebuttal of the authors are highly appreciated. However, the provided answers are not sufficient to change my original score. *


The paper proposes a new model of fine-grained semantic structure of citational sentences to represent their intent and features. Focusing on the task of modeling citations, the paper aims to capture and harmonize the following aspects of the citations: intent type, direction, objects or concepts and relevant context. The authors present a new way of modeling citation through active and passive roles, for which semantic structuring of citation intents are defined. Using this new semantic structure, it presents the process of data selection and annotation. Eventually they evaluate the created dataset qualitatively and quantitatively.

Quality:
Even if it is a strong point that the authors tried to explain most of the concepts discussed in the paper with examples, the citation intents which are the main part of the proposed model are poorly presented.

Novelty:
Intermediate: The novelty of the model they proposed is the fine-grained semantic structures lying behind citational sentences by adapting intents from current works.

Originality:
Good: Most of the intents/classes are adapted from existing works and they are extended with the concepts of objects and context.  Its main contribution is the newly created dataset. Thus, the paper qualifies more as a resource paper than as a research paper.

Significance of this work:
Since the ontology in their framework of the proposed model has not been explained, the significance of the work for the ESWC community is not clear.


**Anonymity:**

Yes, I would like my review to remain anonymous.

**Rating:**

-1: Weak Reject

**Reuse And Availability:**

2: Low

**Strong Points:**

- The research problem has been well motivated using examples.
- The authors have indicated a direction on how to capture the semantics in citational sentences and represent them in structured form.
- As compared to current methods, the proposed model does not include the generic class “background” as it does not provide any useful semantics.


**Subreviewer:**

I delegated this review to a subreviewer.

**Weak Points:**

- The semantic model which is the main contribution of the paper, considers the citing paper as a container in case of the ‘passive role’. This is a disadvantage as it does not capture the relationship that exists between the citing and the cited papers.
- In 3.2, it claims the proposed model creates a knowledge graph instead of a simpler network of articles. However, there are neither examples nor definitions for the “knowledge graph” they suggest.
- There is no description given regarding the ontology in Figure 2 which depicts the main framework of the proposed model. However, it is not known what the classes, entities, and other elements are in this ontology. The authors left it for the user to make assumptions such as the ontology containing the ‘objects’ that occur in citation sentences.   Moreover, it is not clear by just reading the paper what has already been done and what is left for future work regarding the knowledge graph creation process. Since the creation & role of the ontology used has not been clearly explained, it is questionable how it actually fits any of the conference topics.
- In 4.3, the paper explains that they employed a keyword-based random search for “analyze” and “propose”, however, no list of keywords are provided.
- In 4.3, they defined an ordered list of the classes that are supposed to disambiguate ambiguous cases, however, instead of two ambiguous examples for “use” and “propose”, there are no explicit explanations to justify the ordered list, nor any clear examples to support the list.
- The authors have not given any attention to the generalizability of their proposed semantic model beyond the current scope of computer science related scientific papers. The model is not generalizable and it covers a narrow scope in terms of domain of articles. If we use the same model for another dataset with articles from other research areas, the classes defined in this model won’t be effective in representing the citation intents (as e.g. for physical experiments in natural science, theoretical proofs as in mathematics, or discourse structure as in the humanities). This makes the model more or less data-dependent.
- The authors have represented the definitions of the classes using some formalism which is unclear and hard to refer it to as “formal”.  For instance, the usage of logical symbols “implication” is not justified or explained. Besides, the implication for the ‘propose’ class is a single arrow and for the rest it is a double arrow.
- The interpretation of the ‘Analyse’ class causes confusion as the class’s definition does not go along with the example given on page 9 - ‘(A-subject example) “Surveys have been conducted to discuss Big Data Frameworks [CIT].”’ From the example it looks like the subject shouldn’t not be A (the citing paper) but those unknown survey papers which have done the actual analysis.
- Not enough statistics have been presented for the created dataset. It would be informative to mention the minimum and average number of words in the citation sentences and specifically in the contexts.
- There are multiple cases of conceptual ambiguities, self-contradictions and the evaluation using a downstream task is not quite sufficient.

---

> ### Author Rebuttal · Authors · 2021-01-29
>
> We have carefully read all the observations raised by the reviewer and we believe that the review has been affected by some misunderstanding. We try to clarify the most controversial aspects.
>
> [R=Reviewer, A=Answer]
>
> R: The semantic model which is the main contribution of the paper, considers the citing paper as a container in case of the ‘passive role’. This is a disadvantage as it does not capture the relationship that exists between the citing and the cited papers.
>
> A: With respect to the current state of the art, our model introduces additional fine-grained information without losing any relationship. In the specific case of the passive role, the relation between citing and cited papers is maintained and embedded even in the proposed passive-role citation.
>
>
> R: In 3.2, it claims the proposed model creates a knowledge graph instead of a simpler network of articles. However, there are neither examples nor definitions for the “knowledge graph” they suggest. […] Moreover, it is not clear by just reading the paper what has already been done and what is left for future work regarding the knowledge graph creation process.
>
> A: For knowledge graph we mean a semantically-enriched network with typed relations between papers, authors, topics etc. This differs from a simple network of articles created from the author/citation relationships. The definition of the knowledge graph, based on our model, is out of the scope of this paper, as it will be our first future effort.
>
>
> R: There is no description given regarding the ontology in Figure 2 which depicts the main framework of the proposed model.
>
> A: With “ontology” we meant the knowledge graph mentioned above. The aim of Figure 2 was to provide an overview of the framework, including future developments. We will update the description, substituting the term “ontology” with “knowledge graph”.
>
>
> R: In 4.3, the paper explains that they employed a keyword-based random search for “analyze” and “propose”, however, no list of keywords are provided.
>
> A: We will better detail this point in case of acceptance.
>
>
> R: In 4.3, they defined an ordered list of the classes that are supposed to disambiguate ambiguous cases, however, instead of two ambiguous examples for “use” and “propose”, there are no explicit explanations to justify the ordered list, nor any clear examples to support the list.
>
> A: We defined the ordered list as a guideline in order to simplify the disambiguation process.  Our convention is based on sorting the classes from the most specific to the most generic one. To the best of our knowledge, there are no previous works containing specific indications to address ambiguities. We decided to define the mentioned ordered list with the objective of reducing the annotators effort.
>
>
> R: The authors have not given any attention to the generalizability of their proposed semantic model beyond the current scope of computer science.
>
> A: The developed framework can be generally applied to scholarly data where citations play a crucial role within the definition and the extension of research works. Our model can be adapted also to other domains, such as biomedical research, physics, etc., since the proposed features (e.g. objects, roles, etc.) are domain independent. However, we agree that each domain could need further efforts in order to extract the specific entities.
>
>
> R: The authors have represented the definitions of the classes using some formalism which is unclear and hard to refer it to as “formal”. For instance, the usage of logical symbols “implication” is not justified or explained. Besides, the implication for the ‘propose’ class is a single arrow and for the rest it is a double arrow.
>
> A: As regards the chosen formalism, the intention was to explain the rules that define the annotation process and therefore the semantics of the classes. We agree that there are some aspects that we need to uniform, in particular the use of single and double-headed arrows, that represent the citational links between papers. We will revise the definitions towards a more coherent formal representation.
>
>
> R: The interpretation of the ‘Analyse’ class causes confusion as the class’s definition does not go along with the example given on page 9 - ‘(A-subject example) “Surveys have been [...]”’ From the example it looks like the subject shouldn’t not be A (the citing paper) but those unknown survey papers which have done the actual analysis.
>
> A: We agree that this particular example could be misleading, therefore we will update it with a clearer sentence.
>
>
> R: Not enough statistics have been presented for the created dataset.
>
> A: Thank you for the suggestion, we will update the paper with such additional statistics.
>
>
> R: There are multiple cases of conceptual ambiguities, self-contradictions and the evaluation using a downstream task is not quite sufficient.
>
> A: Unfortunately we were not able to link these problems to specific portions of the paper.

---

### Official Review · AnonReviewer1 · 2021-01-13
**More expressive representation for citations than existing work; active/passive role of intents seems to be an original and sensible addition.**

**Rating:** 1
**Confidence:** 2
**Impact:** 3
**Design And Technical Quality:** 3

**Review:**

The authors of this work introduce “CiTelling”, a new model for semantic structures in citational sentences which represents intents and features (i.e., objects, such as quoted methods and additional context). The main contribution is the introduction of their model, which adds more depth to the semantic representation of citations and is thus able to represent citation semantics more accurately than existing representations. A key feature of their representation is that the citing paper A (as opposed to the cited paper B) can have an active or a passive role in a citation. More specifically, when the intent of a citation in work A is to state that some method X has been used, the cited paper B can either be the work that used the method or A itself could be the work that uses the method from B. Hence, this allows to represent cases, which apparently were not possible in the cited previous representation models. Further features of their representation are five intents of a citation (propose, use, extend, analyze, compare) as well as sub-intents where applicable (e.g., use:datasets or use:results). The intents are selected based on previous work but with changes where the authors saw problems in existing representations. Additionally, their model also includes fields for objects and context with meaning depending on the intent. The authors evaluate their model and provide a data set of annotated citations.

Altogether, the differentiation between active and passive citation roles appears to be a useful addition that should drive the field forwards and has not been made before. The selected intents make sense to me, however, the previously existing ones do as well and I’m not able to say whether they are strong improvements and how established the intents from previous works are. The arguments the authors make for their changes, however seem valid to me. One issue I see here, is that occasionally several intents may apply to a citation. The authors added a linear order of intent specificity to solve this where the most specific one that applies should be used. Nonetheless, this shows that some room for confusion is left with this model. The evaluation, while leaving points to improve, seems to be mostly sufficient for a first introduction of the method. Distributions of the different roles and their subclasses are analyzed. Agreement between human annotators is determined for intent classes as well as objects and contexts (through bleu-scores). The authors also evaluate how well a simple machine learning method is able to classify intents and their (active or passive) role in comparison to previous methods. While the comparison here is hard because different classes and models were used, the aim was probably to show that the specific classes make sense and how well it is possible to extract them (semi-)automatically. I think here, the aim of the classification should have been made a bit more clear. The related work, seems to be covered and up to date, although I’m not very familiar with the field.

Some issues I see are concerned with the overall and the formal presentation. While each point is minor on its own, these issues add up to the overall impression. In particular, in the section about citation intents, “formal” representations are given. Although  it is possible to understand what the authors want to express, the used formalism is not known to me and their precise meanings sometimes not clear. The representation does not appear to be consistent (e.g., arrows are single or double-headed without any obvious difference in meaning). In particular, this holds for the symmetric arrow in the compare class. It would be useful to be more precise in the meaning here or, if the formalism is common in some field provide a reference or explanation. Further issues in presentation arise, e.g., in headings of subsections, where words are capitalized sometimes arbitrarily and inconsistently, and in the way subsections and paragraphs are organized. Vector instead of raster graphics should be used wherever possible. Finally, I think for the evaluation an interesting point left out here would be how intents are actually distributed across papers since the data set here is intentionally made to have uniform distribution. I also wonder whether intents cover all kinds of citation sentences found in the literature, since the work mentions that “noise” sentences are filtered out. Still, each of these points on its own is not a major issue and my overall impression is that this work is a useful contribution to the field.



**Anonymity:**

Yes, I would like my review to remain anonymous.

**Reuse And Availability:**

4: High

**Strong Points:**

- More expressive representation for citations than existing work.
- Active/passive role of intents seems to be an original and sensible addition.
- Addition of object and context appears useful to me as well.
- Release of a new data set of annotated citations.
- New selection/definition of intents and sub-intents.

**Subreviewer:**

I delegated this review to a subreviewer.

**Weak Points:**

- Some issues in the formal representation of intents.
- Evaluation could add some minor points.
- Clarity of intentions behind evaluation could occasionally be improved.
- Minor issues in presentation quality.
- Intents leave some minor room for confusion. Might still be better than previous ones.

---

> ### Author Rebuttal · Authors · 2021-01-29
>
> We thank the reviewer for all the words of appreciation and constructive feedback. In case of acceptance, we will pay more attention to formalism and explanations.
>
> We take the occasion to clarify the contribution of our model with respect to the existing literature. First of all, we added novel citation intents defining their semantics formally. This differs from current models that make use of generic classes such as “background”, with very limited or unexpressed meaning. In our work, we proposed a model that enhances the expressiveness of citation modeling with a more fine-grained set of intents.
>
> As regards the chosen formalism, the intention was to explain the rules that define the annotation process and therefore the semantics of the classes. We agree that there are some aspects that we need to uniform, in particular the use of single and double-headed arrows. We will revise the definitions towards a more coherent formal representation.
>
> Our annotation process followed a two-steps approach. In the first phase, the use of currently available classifiers and keyword-based search (with regular expression) produced a noisy dataset (i.e. it contains some erroneous classification). Then, we manually filtered out all incorrect instances. In particular, we did not exclude a priori any kind of citational sentences.

---

### Official Review · AnonReviewer5 · 2021-01-14
**Very interesting work**

**Rating:** 2
**Confidence:** 3
**Impact:** 4
**Design And Technical Quality:** 4

**Review:**

Analysis and modelling of citations in the context of scholarly publications is still a fairly new research domain. This paper addresses some problems existing with ontologies like CiTO which have both an excessive granularity and a shallow specification of the citation structure and context.

I really liked this paper, it is well motivated, written and the proposed citation intents modelling comes together with a preliminary evaluation with neural classifiers.

I have a remark about the choice of producing a balanced dataset: wouldn't this represent unrealistically the distribution of the various citation types? By experience, some types of citations appear rarely.

Another question is about the choice of including critique (from CiTO) as a subclass and not at the same level of the other types (or maybe another way to formulate this question is: what is the advantage of having these classes as subclasses?)

In conclusion, a very interesting work with potential to have an impact on further research in this area and bibliometrics.

**Anonymity:**

Yes, I would like my review to remain anonymous.

**Reuse And Availability:**

4: High

**Strong Points:**

Paper that addresses a relatively new and interesting topic; potential to attract various researchers, especially from bibliometrics. Well written and argumented.

**Subreviewer:**

I submitted this review.

**Weak Points:**

The choice of equally distributed citation intents seems to hide the real distribution of classes

---

> ### Author Rebuttal · Authors · 2021-01-29
>
> We thank the reviewer for all the words of appreciation and constructive feedback.
>
> Regarding the distribution of the classes, our aim was to validate the provided model without any bias coming from the possibile rarity of one class over the others.

---

### Official Review · AnonReviewer3 · 2021-01-14
**Semantic system for citation labelling**

**Rating:** 2
**Confidence:** 3
**Impact:** 3
**Design And Technical Quality:** 4

**Review:**

This paper presents a new system for semantically labelling citations in scientific papers, adding object and context fields and using a different set of classes from previous work.  The paper is well written and easy to understand, with good explanations of the manual annotation and evaluation as well as of the semantic system itself.

It's not clear to me that work like this is being used in practice, but I can see the value of it.

**Anonymity:**

Yes, I would like my review to remain anonymous.

**Reuse And Availability:**

5: Very High

**Strong Points:**

Manual annotation is explained clearly with IAA.

Evaluation results look good compared with previous work,

Adding object and context fields to the class makes sense and looks useful.

Dataset and code are available on Github (I looked through the repository but did not test it).

**Subreviewer:**

I submitted this review.

**Weak Points:**

With regard to motivation, it is not clear that prior work in the field is actually being used in practical applications.

---

> ### Author Rebuttal · Authors · 2021-01-29
>
> We thank the reviewer for all the words of appreciation and constructive feedback. As briefly reported in the conclusions, the proposed model enables the creation of a knowledge graph where reasoning processes and recent development on neural graph embeddings techniques can represent the core of new research and applications on the bibliographic analysis topic.

---

### Official Review · AnonReviewer4 · 2021-01-17
**Novel model for semantic representation of citations; publicly available dataset. Relevant for using citations to build knowledge graphs.**

**Rating:** 2
**Confidence:** 4
**Impact:** 5
**Design And Technical Quality:** 4

**Review:**

Edited to add: thanks for the rebuttal!

The paper proposes a model of fine-grained semantic structures of citation sentences. The topic is interesting, and the proposed model is novel since the model not only identifies the citation intent but also captures the topic of the citation (e.g. a specific object used from the cited work; or a “passive citation” describing the role played by the cited work). The proposed model also has the potential to be used to construct knowledge graphs.

**Pros:**
1.	The paper is well organized, and the description of the model is relatively clear.
2.	The motivating examples in 3.1 "Semantic structures in citations" is quite clear. The aim of differentiating the cited objects as being associated with the citing paper OR the cited paper. It is also novel: mainstream studies on citation intent have not emphasized this aspect. Consequently, the proposed model captures more information that mainstream studies on citation intent.
3.	Feasibility of identifying citation intent and identifying the subject associating with the cited object looks promising according to this model looks feasible, based on classifier performance.

Cons:
1.	The method for identifying the instances for the citation intent needs further elaboration. The model has five citation intents (*use*, *extend*, *analyze*, *propose*, and *compare*). In particular, the method for identifying the instances of *compare* should be described.
2.	Reproducibility needs improvement in some places. In particular, what keywords were used to identify the *analyze* and *propose* citation intents?
3.	Further justification is needed for decisions around the choice of citation intents when there is overlap (section 4.3). The authors' perception of the "informativeness" of the intents is not fully convincing.
4.	Consider using Kappa statics, which have been commonly adopted in research for measuring inter-rater agreement on citation intent and related topics:
Jha, R., Jbara, A.-A., Qazvinian, V., Radev, D.R.: NLP-driven citation analysis for scientometrics. Nat. Lang. Eng. 23, 93–130 (2017). https://doi.org/10.1017/S1351324915000443
Teufel, S., Siddharthan, A., Tidhar, D.: Automatic classification of citation function. In: Proceedings of the 2006 conference on empirical methods in natural language processing. pp. 103–110. Association for Computational Linguistics (2006)
Otherwise, justify the choice of a weighted average here.
5.	Improve clarity.
- The phrases “A-subject” and “B-subject” were confusing to this reviewer. Improving the understandability/intuitiveness of this distinction would be particularly important. (Similarly the discussion of “container” at the top of page 4 could use even a bit more emphasis.)
- Figure 1’s comparison of active role/A-subj and Passive role/B-subj was not clear enough for me.
- Provide motivation for this, and perhaps some further explanation: “Note that we modeled the object and the context fields in a purely semantic way….”.
- Further explanation for footnote 2 would help this reviewer.
- “extending a work means its use after the application of some changes” – perhaps this belongs in your table as helpful context?
- “classes such as extend and compare are mostly associated with very generic objects…” I still don’t have a clear enough understanding of what “object” means here. Do you mean literally the object of the verb?
- “inter-annotation human agreement” -> “inter-annotator agreement”

Additional todo’s in revision:
1.	Mention the dataset and its URL in the abstract
2.	Be more explicit about background in some places:
- Please add citations in the introduction to be more explicit about which papers you are referring to: "Alternatively, recent works proposed Machine Learning approaches for the automatic classification of sentences containing citations into few classes such as extension, use, etc." (In English, generally, use “work” even if there are multiple papers.)
- Describe CiTO in a sentence for readers who may be less familiar with it; also consider linking to the ontology page (you already cite the paper)
- Consider adding citations/URLs to the ACL-ARC dataset [7] and SciCite on page 3 (from your description of [6] at the top of page 3)
3.	Improve readability of Figure 3
4.	You can go deeper into future work – this is interesting and deserves some space here.
5.	Fix the bibliography
- ArXiv URLs are in some cases as old as 2014 – very likely there are formal published versions. Please cite those instead!
- Check reference formatting and consistency (e.g. #12 has the URL twice; #6 has it once – whereas most of your ACL papers don’t have a URL at all.)
6.	Check the language throughout. Some minor language points:
- In section 2, you use “the authors” in reference to authors of [22] and [12]; consider rephrasing because this can be confused with YOU (the authors of THIS paper)
- “critics” in Table 1 should be “critiques”
- P10 “we found that may exist” -> “we found that there may exist”
- P12 “resulted to be” -> “was”

**Anonymity:**

Yes, I would like my review to remain anonymous.

**Reuse And Availability:**

4: High

**Strong Points:**

* Well-motivated work
* Novel approach to a widely discussed and important task
* Data availability
* Strong results

**Subreviewer:**

I submitted this review.

**Weak Points:**

* Clarity needs improvement
* Reproducibility could be improved (e.g. description of identifying the instances of *analyze*, *propose*, and *compare*)
* Data URL needs to be added to the abstract

---

> ### Author Rebuttal · Authors · 2021-01-29
>
> We thank the reviewer for all the words of appreciation. We will focus our work on the constructive feedback we received. In particular we will improve the clarity of section 3.2 on active and passive roles, including the scheme reported in Figure 1.
> Moreover, we will better detail points such as the keywords used for the initial crawling of citation instances, and other minor issues (data url, bibliography, readability of Figure 3 and the reported typos).

---

### Decision · Program_Chairs · 2021-02-23

**Decision:**

Accept

**Comment:**

The paper proposes a model of fine-grained semantic structures of citation sentences.
From a general perspective the paper is well written and the topic it tackles is interesting. Additionally the solution proposed is sound and novel.
There is a broad consensus among reviewers on the acceptance of the paper. Hence, I recommend to accept the paper.